# Deep Learning Applications in Geosciences: Insights into Ichnological Analysis

**Korhan Ayranci** [1,*] , **Isa E. Yildirim** [2], **Umair bin Waheed** [1] **and James A. MacEachern** [3]

1 Department of Geosciences, College of Petroleum Engineering & Geosciences, King Fahd University of Petroleum & Minerals, Dhahran 31261, Saudi Arabia; umairbin.waheed@kaust.edu.sa
2 Institute of Applied Mathematics, Middle East Technical University, Ankara 06800, Turkey; isa.yildirim@kaust.edu.sa
3 ARISE, Department of Earth Sciences, Simon Fraser University, Burnaby, BC V5A 1S6, Canada; jmaceach@sfu.ca
* Correspondence: ayranci@gmail.com; Tel.: +966-53-140-8025

**Abstract:** Ichnological analysis, particularly assessing bioturbation index, provides critical parameters for characterizing many oil and gas reservoirs. It provides information on reservoir quality, paleodepositional conditions, redox conditions, and more. However, accurately characterizing ichnological characteristics requires long hours of training and practice, and many marine or marginal marine reservoirs require these specialized expertise. This adds more load to geoscientists and may cause distraction, errors, and bias, particularly when continuously logging long sedimentary successions. In order to alleviate this issue, we propose an automated technique to determine the bioturbation index in cores and outcrops by harnessing the capabilities of deep convolutional neural networks (DCNNs) as image classifiers. In order to find a fast and robust solution, we utilize ideas from deep learning. We compiled and labeled a large data set (1303 images) composed of images spanning the full range (BI 0–6) of bioturbation indices. We divided these images into groups based on their bioturbation indices in order to prepare training data for the DCNN. Finally, we analyzed the trained DCNN model on images and obtained high classification accuracies. This is a pioneering work in the field of ichnological analysis, as the current practice is to perform classification tasks manually by experts in the field.

**Keywords:** ichnology; deep learning; transfer learning; artificial intelligence; trace fossils

## 1. Introduction

Machine learning is a subfield of artificial intelligence that aims to learn structure in data and fit those data into models that can be utilized for automating mundane tasks and gaining intelligent insights. It has shown great potential in tackling long-standing research problems across science and engineering disciplines. Most noticeable has been the contributions of deep learning, a branch of machine learning that is founded on artificial neural networks. It has made remarkable breakthroughs in a variety of fields, including biology, natural language processing, and computer vision [1–5].

In recent years, deep learning has attracted growing attention from the geoscience community, as it yields fast and accurate solutions for labor-intensive research in areas such as petroleum exploration, heavy mineral analysis, facies analysis, and the monitoring of volcanoes using complex algorithms e.g., [6–8]. For example, image classification using convolution neural networks (CNNs) has recently shown remarkable levels of performance in core-based facies analysis e.g., [9], where long hours of visual observation are essential. Some of the advantages of using deep learning in core-based studies are: (1) reducing risks of human error by shortening physical labor; and (2) highlighting critical or problematic core intervals, and thereby focusing human attention more efficiently.

Although facies analysis is crucial, ichnological analysis is equally important to geo-scientists in determining paleodepositional environments. Unfortunately, it also requires long hours of training, practice, and visual observation. Therefore, ichnological analysis is predisposed to benefiting from deep learning algorithms. There are various ways of applying ichnological analysis in core-based datasets, including identification of individual trace fossils, assessing trace fossil assemblages, calculating bioturbation index, recording the diversity of trace fossil suites, and evaluating burrow size distributions [10–17]. Among these, bioturbation index, expressed as Bioturbation Index (BI), is perhaps one of the most fundamental criteria for understanding paleodepositional conditions, particularly with respect to recognizing the presence of physico-chemical stress factors that operate in the paleoenvironments deposited in various hydrocarbon reservoir units.

There are six grades of bioturbation intensities encompassed by the original BI definition: BI 0–6. BI 0 characterizes facies with no visible bioturbation, whereas BI 6 indicates complete biogenic homogenization of the media, wherein no preserved primary physical sedimentary structures survive, and all sedimentary fabric is biologically induced. Although BI is technically a quantitative index [18], determining precise values is challenging in practice. Correspondingly, many studies tend to employ alternative techniques, such as using rough percentage values and visually defining low/moderate/intense bioturbation indices, instead of the six grades of bioturbation in the BI classification.

In order to find a more robust, fast, and consistent solution to the problem of bioturbation index classification, we employed the capabilities of DCNNs. We compiled a large data set (*n* = 1303) composed of images with different bioturbation indices. We first divided these images into two groups to test whether the proposed image classification technique can aid in differentiating unbioturbated facies from bioturbated facies. This approach can be critical in unconventional reservoirs, where unbioturbated units, possibly suggesting anoxic conditions, tend to have higher total organic carbon preservation compared to bioturbated units; however, not all unbioturbated muddy units are indicative of oxygen-depleted conditions [19,20]. We then used the original bioturbation index values and divided images into three classifications: unbioturbated (0%), moderately bioturbated (1–30%), and intensely bioturbated (31–100%), corresponding to BI 0, BI 1–2, and BI 3–6, respectively. To the best of our knowledge, this is the first study that uses deep learning-based image classification in determining ichnological characterization of facies from core-based (subsurface) datasets. Since DCNNs typically require large amounts of labeled data and long training times for optimal performance, we instead used transfer learning to speed up the training process. Transfer learning is a machine learning technique that allows us to use information gained from solving one problem to solve the next. In this case, we used the VGG-16 [4] network trained on the ImageNet dataset. Based on tests comparing the performance of other popular DCNN architectures, we found VGG-16 to yield the best performance in terms of accuracy and computational efficiency for the ichnology classification task.

Although image classification, at its current stage, cannot replace human interpretation, it promises tremendous advantages, including: (1) reducing the amount of time spent in the logging of cores; (2) minimizing the risk of human error; and (3) reducing costs by using direct human attention more efficiently.

The rest of the paper is organized as follows. We begin with a description of the dataset and the DCNN architecture used. This is followed by a discussion of the results and a brief note on potential future applications, before mentioning the conclusions and implications of this study.

## 2. Materials and Methods

### 2.1. Data Set

The images used in our experiments were collected from a variety of subsurface cores and outcrop exposures representing siliciclastic sedimentary facies from several Cretaceous-aged stratigraphic formations in the Western Canada Sedimentary Basin, Alberta Canada.

In our training data set, there was a large variation in individual trace fossils present in the facies, including *Asterosoma*, *Chondrites*, *Conichnus*, *Cylindrichnus*, *Diplocraterion*, *Macaronichnus*, *Ophiomorpha*, *Palaeophycus*, *Planolites*, *Phycosiphon*, *Piscichnus*, *Rhizocorallium*, *Rossellia*, *Schaubcylindrichnus*, *Siphonichnus*, *Skolithos*, *Teichichnus*, and *Thalassinoides*, as well as bio-deformation structures and escape structures.

The majority of the images were derived from sandstone and siltstone facies with some mudstone intervals, as well as rare conglomerate beds and mud-clast breccia units. All images were derived from facies recording relatively shallow-water settings, such as estuaries, bays, shorefaces, offshore-shelf, delta fronts, and prodeltas. A wide range of physical sedimentary structures was observed, including low-angle to horizontal parallel lamination, high-angle cross-stratification, trough cross-stratification, hummocky cross-stratification, current ripple lamination, oscillation ripple lamination, normally graded bedding, fluid mud drapes, gravel lags, isolated clasts and rip-up clasts, loading and flame structures, convolute bedding and other soft-sediment deformation structures, syneresis cracks, and various types of concretions.

### 2.2. Network Architecture

Among popular computer vision tasks, such as object detection, image segmentation, and image-resolution enhancement, image classification is one of the most addressed application areas using deep convolutional neural networks (DCNNs) [1–5]. DCNN is a class of deep networks formed by a series of interconnected neurons that has led to great achievements on image classification problems. For example, Krizhevsky et al. [1] proposed AlexNet, a DCNN designed to classify images from the ImageNet dataset, which contains 1.2 million high-resolution human-annotated images with 1000 different classes. Later, Simonyan and Zisserman [4] introduced a deeper architecture for better classification accuracies for the ImageNet dataset. Their proposed architectures, namely the VGG networks (e.g., VGG-16 and VGG-19), could also be applied to other image recognition tasks.

For the bioturbation classification problem, we propose to train a DCNN using BI data labeled by an experienced ichnologist. These deep networks contain a large number of trainable parameters requiring a huge amount of labeled data. To avoid the need for manually labeling a massive collection of training examples, we used transfer learning—a deep learning technique that relies on storing knowledge gained while solving one problem and then applying it to a different but related problem. Therefore, instead of training from scratch, we rely on the VGG-16 convolutional neural network (CNN) model [4] and use part of its learned parameters derived from the training on the ImageNet dataset. This approach allows us to gain the same high classification performance with a minimal amount of manual labeling required. Such an approach has been proven to be successful in prior studies on a diverse set of applications [9,21–25].

Our DL approach belongs to the supervised learning class, which aims at learning a function from known input and output pairs (training data). The learned function is then used in the prediction stage to map an unseen input (image) to an output (bioturbation index class). Teaching a DCNN to learn this mapping involves data preparation, model definition, and training. We used a total of 1303 color images that were selected approximately between 4 to 8 cm in size in order to make our algorithm applicable to core-based studies. Our outcrop images were likewise cropped roughly in the size of core dimensions. As the default input size for the pre-trained model, we used $224 \times 224$ pixels, and we resized our images to the same size while visually preserving original sedimentary and ichnological structures without distortion. We carefully identified and labeled bioturbation indices in 3 classes. Among the 1303 images, 530 of them belonged to BI 0, 360 of them represented BI 1–2, and 413 of them represented BI 3–6. From these images, 1041 were used for training (79.9% of the overall images) and the remaining 262 (20.1% of the overall images) were used as test data for the evaluation and prediction stages.

Perhaps one of the most difficult tasks in such classifications is to identify images that encompass more than one facies possessing different bioturbation indices (e.g., Figure 1). For such cases, we took average BI values of each bed and their thicknesses into consideration, particularly for the ones that were very close to the lower-end or upper-end cut-off values. Between BI 1–2 and BI 3–6, 74 images revealed this issue, and their bioturbation indices were calculated using the following formula:

$$BI_{avg} = \frac{(T_a \times BI_a) + (T_b \times BI_b)}{\sum T}$$

where $BI_{avg}$ is the average bioturbation index, $T_{a, b}$, is the thickness of the individual beds, $BI_{a, b}$, is the bioturbation index of each bed, and $\sum T$ is total thickness.

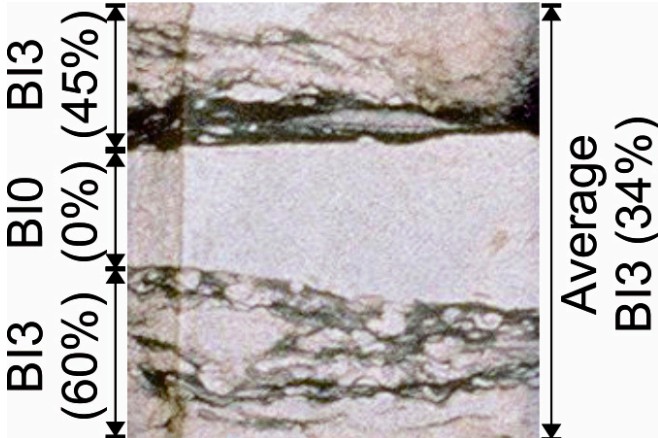

**Figure 1.** A representative core image showing various bioturbation indices expressed by bioturbation index (BI) in different facies.

We carried out two main experiments in this study. First, we trained a DCNN to differentiate between bioturbated and unbioturbated facies. Next, we extended this task to a 3-class classification problem that sough to model map the input images to one of the three classes of unbioturbated (BI 0), moderately bioturbated (BI 1–2), or intensely bioturbated (BI 3–6) facies. For both experiments, we defined our models (Figure 2) based on the VGG-16 architecture [4], which was a DCNN containing 16 layers composed of $3 \times 3$ convolution filters. The input of the network was a $224 \times 224$-pixel image with 3 color channels (red, green, and blue). From the given inputs, their low-level features (lines, edges, or dots) were learned by the convolution filters in the shallow parts of the network. In the deeper parts, high-level features such as objects were more detectable. This is the reason why feature extraction is more problem-specific in the deeper levels of the network. Therefore, we kept the parameters fixed for the first four blocks from the training of the VGG-16 for ImageNet dataset, and only allowed the weights to be trained for the last block and the classifier parts of our models. Thus, using transfer learning significantly reduced the computation time needed to train the model, and reduced the need for a large number of training examples, as only the last block and the fully connected layers had trainable parameters.

It is worth noting that through tests involving popular DCNN architectures such as VGG-16, VGG-19, and custom DCNN models, we found VGG-16 to yield the best classification performance. Moreover, starting with pre-trained weights for the ImageNet dataset, we tested different combinations of frozen and trainable blocks. We found that freezing the first four blocks of VGG-16, while allowing the last block and the fully connected layers to be trainable, resulted in efficient training and improved classification accuracy compared to other configurations.

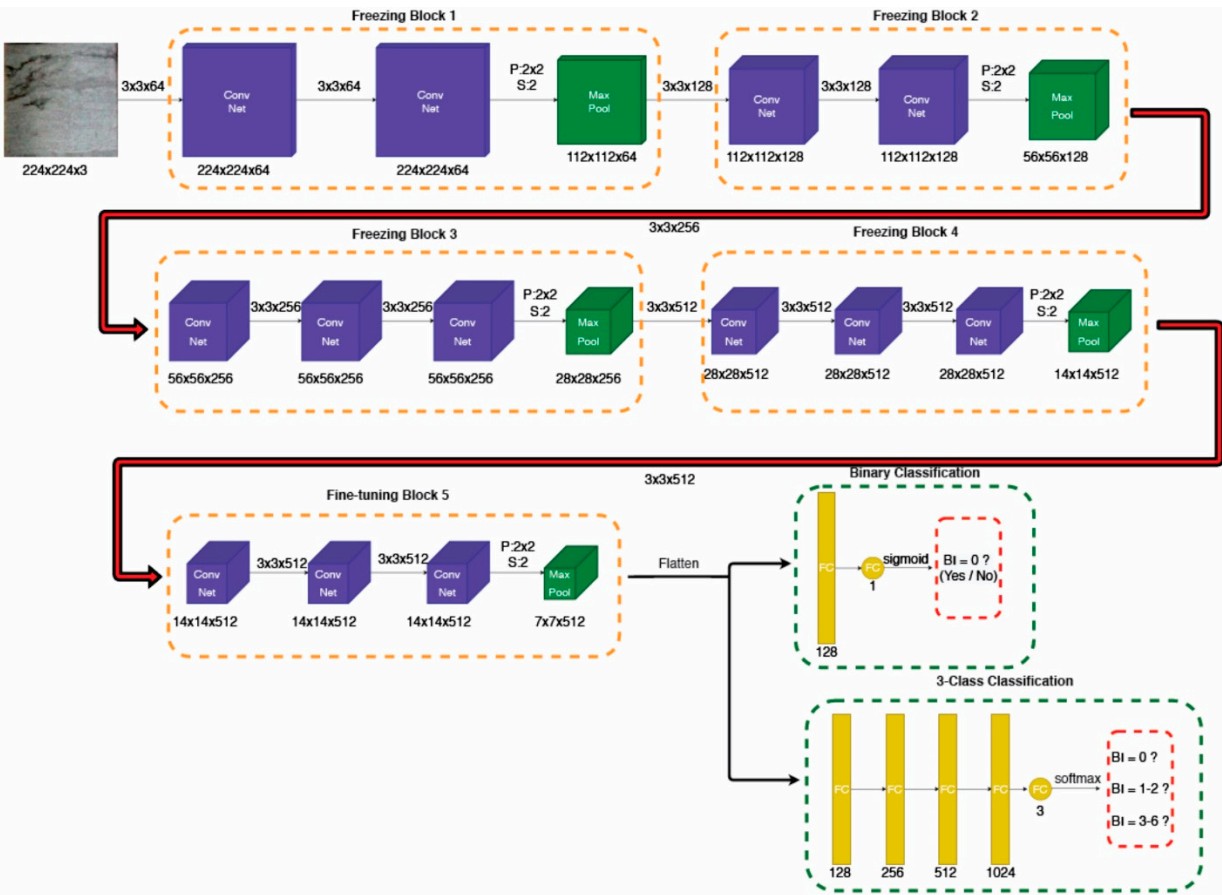

**Figure 2.** A convolutional neural network architecture based on VGG-16 with fully connected layers at the end. The architecture describes the routines for both binary and multi-class classifications. The input is a three-channel (red, green, blue) image. The binary classification problem outputs whether the input image is bioturbated or not. The multi-class classification problem outputs the class of the given image in terms of its bioturbation index.

Moreover, to expand the existing labeled data and to insert regularizing effects into the classification problems, we also used data augmentation during the training stage. Data augmentation is a commonly used technique in model training, wherein new samples are artificially created from the training data by applying some basic operations (rotation, cropping, flipping, scaling, shifting, etc.). This allows a network to detect complex internal features more easily when training data are limited. In order to not distort the particular characteristics of the images, we carefully chose the augmentation techniques and allowed the algorithm to create new samples randomly, applying horizontal flip and 10% shifts in the width and height to the training data.

## 3. Results and Discussion

In order to ascertain the applicability of deep learning to ichnological analysis, we ran two main experiments. Following this, we then tested a few specific techniques to eliminate misclassifications and increase the overall accuracy of the algorithm. The first experiment was for determining the presence of bioturbation in images and the second experiment was to classify images further based on three main bioturbation indices.

### 3.1. Experimental Results 1: Unbioturbated versus Bioturbated Facies

The first experiment (EXP#1) is for differentiating unbioturbated facies (BI 0) from bioturbated facies (BI 1–6) (Figure 3). To achieve this, we train our algorithm using BI 0 images and BI 1–6 images (Table 1). The algorithm ran ten times on a test data set comprising 106 BI 0 and 156 BI >0 test images (Table 1). Accuracy values range between

93.8% and 97.7%, with an average accuracy value of 95.9%. The analysis below represents the highest accuracy results.

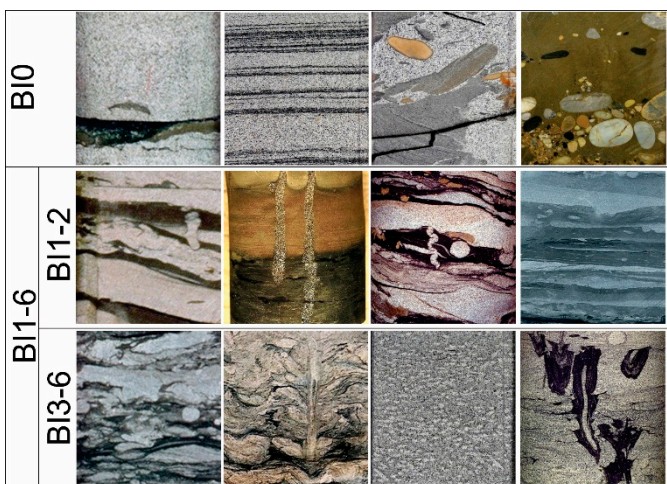

**Figure 3.** Examples of the expert-labeled training images used in our algorithm. Overall, 1303 core and outcrop images are labeled based on their bioturbation indices.

**Table 1.** Results of experiment #1.

| Experiment #1 | | | | | |
|---|---|---|---|---|---|
| | **Total Number of Images** | **Number of Training Images** | **Number of Test Images** | **Accurate** | **Misclassified** |
| BI 0 | 530 | 424 | 106 | 104 (98.1%) | 2 (1.9%) |
| BI 1–6 | 773 | 617 | 156 | 152 (97.4%) | 4 (2.6%) |
| Total | 1303 | 1041 | 262 | 256 (97.7%) | 6 (2.3%) |

Among BI 0 and BI 1–6 test images, 98.1% and 97.4% are correctly classified, respectively. Complex primary structures (e.g., hummocky cross-stratification and amalgamated current ripples) and several non-biogenic structures that resemble biogenic features (e.g., mudstone rip-up clasts, concretions, gravel lags, scattered coal fragments, flame structures, fractures/cracks, and soft-sediment deformation) are correctly classified or did not cause confusion for the network. The algorithm also successfully classified outcrop images along with core images and facies characterized by various grain sizes (e.g., fine-grained sand to gravel), and successfully ignored human-made features such as pen marks and surface stains.

In order to better understand our results and improve our algorithm for future applications, we also analyzed the misclassified images and compared them with similar images in the training data set. Only 1.9% of the BI 0 test images are misclassified as BI 1–6, and 2.6% of the BI 1–6 test images are misclassified as BI 0 (Table 1). These misclassified images display either rare, deformed mud drapes resembling bioturbation (e.g., Figure 4A), extremely diminutive trace fossils, individual ichnogenera for which there are a few or no examples in the training data set (Figure 4B), or sediment-swimming structures for which there are no images in the training data set.

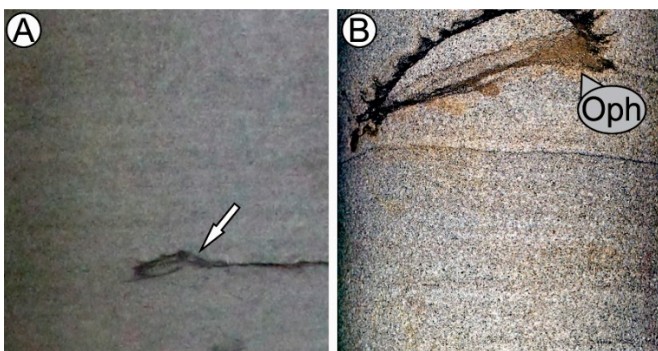

**Figure 4.** Representative misclassified images in EXP#1: (**A**) image showing a deformed mud-drape (white arrow). The deformation caused the mud-drape to resemble bioturbation labeled as BI 1–2 present in some of the other images in the training images. (**B**) Individual *Ophiomorpha*, which is misclassified owing to the lack of similar images in the training dataset.

### 3.2. Experimental Results 2: BI 0 vs. BI 1–2 vs. BI 3–6

Our second experiment (EXP#2) is to test whether we can construct a 3-class bioturbation index classification using the same deep learning model. In this experiment, all images were grouped into three categories: (1) unbioturbated (BI 0), (2) moderately bioturbated (BI 1–2), and (3) intensely bioturbated (BI 3–6). In other words, we divide BI 1–6 images used in EXP#1 into two classes as BI 1–2 and BI 3–6 (Figure 3). We run the algorithm on test data set of 106 BI 0, 73 BI 1–2, and 83 BI 3–6 images. Again, we train our algorithm ten times, starting with different initializations for the trainable parts of the model, and test them on the same test data. The lowest accuracy we obtain is 85.1%, whereas the highest is 88.9%, with an average of 86.8% for the three-class classification problem (Table 2 and Figure 5).

**Table 2.** Results of experiment #2.

| | Experiment #2 | | | | |
|---|---|---|---|---|---|
| | **Total Number of Images** | **Number of Training Images** | **Number of Test Images** | **Accurate** | **Misclassified** |
| BI 0 | 530 | 424 | 106 | 100 (94.3%) | 6 (5.7%) |
| BI 1–2 | 360 | 287 | 73 | 62 (84.9%) | 11 (15.1%) |
| BI 3–6 | 413 | 330 | 83 | 71 (85.5%) | 12 (14.5%) |
| Total | 1303 | 1041 | 262 | 233 (88.9%) | 29 (11.1%) |

Based on the highest test accuracy results (88.9%), the majority of the BI 0 images are correctly classified, similar to the results of EXP#1 (Tables 1 and 2). Given that the BI 0 data set is identical in both experiments, correctly identified images are also similar . Therefore, 94.3% of the BI 0 images were correctly classified (Table 2 and Figure 5). However, EXP#2 showed slightly different accuracy results when the BI 1–6 class was divided into BI 1–2 and BI 3–6 classes. Among the 73 BI 1–2 test images, 84.9% are correctly classified. From the 83 BI 3–6 test images, 85.5% are correctly classified.

Only 5.7% of the BI 0 images are misclassified in EXP#2, with most of these either identical to or similar to the images that are misclassified in EXP#1. EXP#2 shows lower precision in BI 1–2 images, with 15.1% misclassified. The vast majority of these misclassified images are labeled as BI 0 in our original classification. These images show very low bioturbation indices (the lower range of BI 1), large concretions, and single isolated trace fossils. For example, two misclassified images display diminutive escape structures expressed only by minor disruptions of laminae (Figure 6A), unlike the more pronounced ones included in the training data set. One image shows a large-scale escape structure but also includes some mineral formations or possible smeared burrows (e.g., Cylindrichnus) (Figure 6B). Precision in predicting BI 3–6 images is slightly higher compared to BI 1–2 images, with only 14.5% misclassified. The majority of these misclassified images occur at

the lower limits of BI 3 (e.g., approximately 31–45%) and are thus intergradational with the upper limits of BI 2.

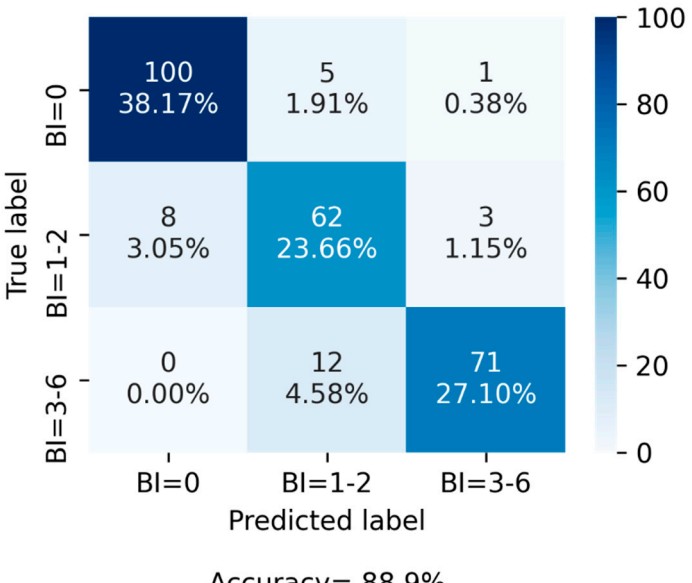

Accuracy= 88.9%

**Figure 5.** Confusion matrix for Experiment #2 indicating classification performance for the three bioturbation classes.

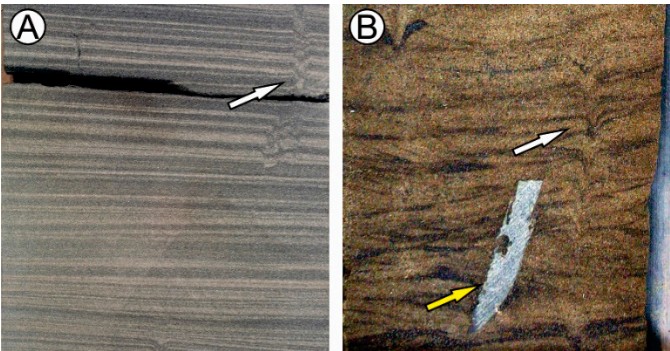

**Figure 6.** Two images as examples of misclassified images in EXP#2: (**A**) images displaying parallel lamination with a small escape structure (white arrow). (**B**) Image showing large escape structure (white arrow) along with a burrow or mineral growth (yellow arrow).

It should be noted that after EXP#1, but before the final results of EXP#2 were obtained, we used 877 training images for both experiments. Although initial EXP#1 results showed high accuracies, initial EXP#2 results showed considerably lower accuracies than the final EXP#2 results explained above. This issue was the lack of statistical diversity of the training dataset after we divided BI 1–6 images into two subclasses in EXP#2. To eliminate this issue, we added additional 82 images into the BI 1–2 class and 82 images into BI 3–6 class and ran the algorithm 10 times with different seeds. The final results of our experiments (EXP#1 and EXP#2), which are presented herein (Tables 1 and 2), include these additional images.

## 4. Future Applications

Before and after compiling the EXP#1 and EXP#2 results, we tested two techniques to improve algorithm accuracy. The first one was to test what could be achieved on specific features that are routinely misclassified. One particular misclassified image in EXP#2 shows a well-defined Ophiomorpha (Figure 4B). Upon analyzing the training data set, we realized that it did not contain any similar images. Thus, we specifically added two

more images containing examples of Ophiomorpha into our training data set and ran the experiment again. Results showed clear improvement, in that this specific trace fossil and the image were then correctly classified. The second test was for the entire dataset accuracy, and it was applied before the final results of EXP#2 were received. Initially, we received low accuracy values for EXP#2, but then we added more images into the training data set. This increased the overall accuracies considerably (around 4–5%), resulting in higher accuracies (88.9%) and suggesting that adding more training data with statistical diversity allows improved accuracies in our model.

It should be noted that pure mudstones, such as oil and gas shales associated with unconventional hydrocarbon plays, display significantly different ichnological characteristics [19]. Correspondingly, facies examples of such unconventional shale reservoirs are not included in this study. Although we tested our algorithm using shale images (e.g., the Horn River shales), owing to the diminutive nature of the trace fossils, we achieved low accuracy results. Such settings should probably be treated separately.

## 5. Conclusions and Implications

Applications of deep learning to ichnological studies are documented here for the first time. Our results clearly show high accuracies in predicting bioturbation indices in various siliciclastic sedimentary rock formations. These applications can be vital in oil and gas exploration through reducing uncertainty, lowering the cost and labor time of experts, maximizing efficiency by directing the experts' attention to more problematic intervals (i.e., those yielding low accuracy results), and in academia by facilitating accurate, reliable, comparable, and consistent paleoenvironmental interpretations.

Our algorithm can be applied to industry and academia as long as image dimensions are roughly comparable to those observed in subsurface cores. Therefore, it can be used by any geologist with limited or no ichnological training. One needs to ensure that the image dimension is consistent with the input size of the network ($224 \times 224$ pixels) and the resolution of the image is consistent with that used in training the model. We used images with approximately 6–8 cm width and height, resulting in a resolution of 28–37 pixels/cm.

The algorithm is also suitable for both core- and outcrop-based studies. With such an application, ichnology can be utilized widely worldwide, and geoscientists can achieve a better understanding of the ichnological characteristics of siliciclastic reservoir units. Applications of automated ichnology in various formations can also provide more accurate and correlatable results between different formations.

Perhaps one of the most interesting findings in our algorithm is that, even should some features resemble trace fossils, such as rip-up clasts and concretions, they were mostly correctly classified as non-trace fossils in our tests. This may be due to their shape, internal structure, and the marked contrast between them and the surrounding media. For non-ichnologists in particular, these features can be misleading; therefore, our algorithm can yield superior results in these cases.

From a deep learning perspective, our experiments reveal that the number of expert-labeled images in the training dataset plays an important role in achieving high accuracies. In an earlier attempt, we used 1141 images and achieved 94.7% and 85.0% accuracies in EXP#1 and EXP#2, respectively. Given that the EXP#2 accuracy was initially relatively low, we introduced 162 more images, which led to a 3.0% and 3.9% increase in EXP#1 and EXP#2, respectively. With further additions to the training set, higher accuracies can be expected.

Geosciences require significant numbers of visual observations in many different sub-disciplines, such as mineralogy, volcanology, sedimentology, and petroleum geology. Therefore, deep learning offers a wide range of geological applications. In the future, individual trace fossil identifications and detailed facies identifications will be implemented in our dataset, providing not only accurate assessments of bioturbation index but a comprehensive core-based analysis.

**Author Contributions:** Conceptualization, K.A. and U.b.W.; methodology, U.b.W. and I.E.Y.; resources, J.A.M.; data curation, K.A. and J.A.M.; writing—original draft preparation, K.A., I.E.Y. U.b.W. and J.A.M.; writing—review and editing, K.A., I.E.Y., U.b.W. and J.A.M. All authors have read and agreed to the published version of the manuscript.

**Funding:** This research received no external funding.

**Acknowledgments:** Funding to J.A.M. is through a Natural Sciences and Engineering Research Council (NSERC) Discovery Grant, and is gratefully acknowledged. The study was supported by start-up grants from the College of Petroleum and Geosciences, KFUPM.

**Conflicts of Interest:** The authors declare no conflict of interest.

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
