# Peer review of "Deep Learning Applications in Geosciences: Insights into Ichnological Analysis"

_applsci, doi:10.3390/app11167736_

Round 1
Reviewer 1 Report
The proposed manuscript introduces the use of Deep Learning paradigm into the ichnological analysis for automatizing the process and making it more robust with respect to human errors.
The paper does not present any particular novelty in general, but it migrates paradigms and architectures used in other fields (i.e., computer vision) for the analysis of fossils image data, which can be considered as small novelty for the ichnology.
The paper is written in a quite good English level, with few grammar mistakes or typos (e.g., line 11). The reviewer would suggest a general revision before the submission also, in order to make the manuscript properly incisive and coherent.
The reviewer would also suggest few points to be addressed for a future publication, described as follows:
1 - A reminder of the manuscript at the end of Section 1 would be recommended for the reader.
2 - A general revision of the manuscript needs to be conducted for highlighting the state of the art and the niche covered by the proposed contribution in Sec.1, as well as for expanding the content of the contribution in the following sections.
3 - Quantitative evaluations (i.e., line 56) are suggested to be moved away from Sec.1.
4 - According to the reviewer, a separation of Sec. 2 between the both the data and network architecture considered and the processing conducted would be beneficial to the reader, with a more clear and incisive illustration of the corresponding parts.
5 - During the analysis, did you perform any cross-validation algorithm to check the robustness with respect to any change in the training set (e.g., k-fold CV)?
6 - Considering line 150, have you considered keeping the weights of some layers more in the last block of the net (i.e., training only the last layer and the final classifier)? Would be this beneficial for your application? Please, clarify this point.
7 - The experimental analysis lacks in performance comparison between the proposed methodology and the state-of-the-art (see Table 1). Moreover, the gathering of validation and training samples in the accuracy performance in Table 1 can be misleading or at least pointless.
Please, consider the separate analysis of training and validation.
8 - In general, by considering the motivations asserted in your paper related to task automating, I do not clearly see the motivation of working with DCNN architectures. Were other automatic ML techniques (either supervised or not) previously consider in the field, and if yes, what were their drawbacks?
Do you have any specific motivation of using VGG net instead of other CNN architectures (e.g., Alexnet), possibly with a shallower structure? Please clarify this point.
Author Response
Dear Editor and reviewer #1:
Please find attached, the revised manuscript (Manuscript ID applsci-1308441):
Deep-Learning Applications in Geosciences: Insights into Ichnological Analysis
By: Korhan Ayranci*, Isa E. Yildirim, Umair bin Waheed and James A. MacEachern
We thank the reviewers and editor for their thoughtful comments aimed to improve the quality of the manuscript. We have carefully considered their comments and suggestions, and revised our manuscript accordingly. The following summarizes our responses to the reviewers and editor comments (in red).
Response to comments from reviewer #1
- A reminder of the manuscript at the end of Section 1 would be recommended for the reader.
We have incorporated the suggestion and added text at the end of the Introduction section to remind the reader of the main message from this study and the outline of the rest of the paper.
- A general revision of the manuscript needs to be conducted for highlighting the state of the art and the niche covered by the proposed contribution in Sec.1, as well as for expanding the content of the contribution in the following sections.
Thank you for your comment. We have now added more text at several locations throughout the manuscript to highlight the state of the art. The current practice is to perform the analysis manually and, therefore, this work is first of its kind in the field.
- Quantitative evaluations (i.e., line 56) are suggested to be moved away from Sec.1.
Done. We have removed percentage values of bioturbation indices from line 56. We would like to keep the percentage values of unbioturbated, moderately bioturbated, and intensely bioturbated indices in the following sentence as we believe that these values will provide readers a better understanding of how these indices are different from each other.
- According to the reviewer, a separation of Sec. 2 between the both the data and network architecture considered and the processing conducted would be beneficial to the reader, with a more clear and incisive illustration of the corresponding parts.
We agree with your comment. We have now separated methodology from the data set and arranged the information in separate subsections. We hope this will make it clearer for readers. Thank you for bringing this to our attention
- During the analysis, did you perform any cross-validation algorithm to check the robustness with respect to any change in the training set (e.g., k-fold CV)?
To analyze the robustness of the trained model, we trained the DCNN multiple times starting with different seeds and reported the outcome in Section 3.
- Considering line 150, have you considered keeping the weights of some layers more in the last block of the net (i.e., training only the last layer and the final classifier)? Would be this beneficial for your application? Please, clarify this point.
Indeed, we studied this aspect through multiple experiments and found the proposed combination to be the optimal. We have added some text to clarify this point for readers.
- The experimental analysis lacks in performance comparison between the proposed methodology and the state-of-the-art (see Table 1). Moreover, the gathering of validation and training samples in the accuracy performance in Table 1 can be misleading or at least pointless. Please, consider the separate analysis of training and validation.
The state of the art in the field of ichnological analysis is to perform the task manually/visually. To the best of our knowledge, this is the first attempt that uses an algorithm to automate the task. Therefore, we consider this work to be a paradigm shift in this field. We have clarified this point throughout the text for readers.
We agree with your comment regarding Table 1. We have now removed percentage values from the number of training and test images, which makes things clearer for the reader.
- In general, by considering the motivations asserted in your paper related to task automating, I do not clearly see the motivation of working with DCNN architectures. Were other automatic ML techniques (either supervised or not) previously consider in the field, and if yes, what were their drawbacks? Do you have any specific motivation of using VGG net instead of other CNN architectures (e.g., Alexnet), possibly with a shallower structure? Please clarify this point.
This is the first attempt in automating the ichnological classification problem. Since DCNNs are designed to mimic human vision, therefore, they are well-suited to the task. We tried different popular DCNN architectures and found VGG-16 with the proposed network configuration to yield the optimal performance. We have updated the text to reflect these ideas.
Sincerely,
Dr. Korhan Ayranci

Reviewer 2 Report
The paper is interesting but should be improved in some parts:
- A section relating to the state of the art is completely absent;
- The method should be better described highlighting the innovative aspect;
- The experimental section does not present comparisons with methods that work in the same field.
- In the same context, with regard to DCNNs, the following paper should be mentioned:
Manzo, M., & Pellino, S. (2021). Fighting together against the pandemic: learning multiple models on tomography images for COVID-19 diagnosis. AI, 2 (2), 261-273.
Author Response
Dear Editor and reviewer #2:
Please find attached, the revised manuscript (Manuscript ID applsci-1308441):
Deep-Learning Applications in Geosciences: Insights into Ichnological Analysis
By: Korhan Ayranci*, Isa E. Yildirim, Umair bin Waheed and James A. MacEachern
We thank the reviewers and editor for their thoughtful comments aimed to improve the quality of the manuscript. We have carefully considered their comments and suggestions, and revised our manuscript accordingly. The following summarizes our responses to the reviewers and editor comments (in red).
Response to comments from reviewer #2
- A section relating to the state of the art is completely absent.
The current practice is that an expert geologist in the field of ichnology performs the task manually based on expertise accrued over the years. We introduce a paradigm-shift to the field of ichnological analysis and to the best of our knowledge, this is the first study that proposes an automated solution to the problem. We have added text at different locations in the text to clarify this point further.
- The method should be better described highlighting the innovative aspect.
We have added some text in the Introduction to highlight the innovative aspect of the paper.
- The experimental section does not present comparisons with methods that work in the same field.
Thank you for your comment. We have emphasized the point that this is the first study that automates the task of ichnological analysis and as such there are no other algorithms to compare the performance with.
- In the same context, with regard to DCNNs, the following paper should be mentioned:
Manzo, M., & Pellino, S. (2021). Fighting together against the pandemic: learning multiple models on tomography images for COVID-19 diagnosis. AI, 2 (2), 261-273.
Thanks for the suggestion, We added the paper as a reference to the DCNN applications.
Sincerely,
Dr. Korhan Ayranci

Round 2
Reviewer 1 Report
The manuscript presents relevant improvements with respect to the previous version and the authors addressed most of the points. The manuscript presents only few minor points to be addressed for a clear reading and comprehension.
1) In Sec.2.1 a good qualitative description of the dataset from ichnological perspective is provided; however, the reivewer would suggest to illustrate also the number of images considered for the different categories, whether grayscale or colour, and in which proportions for the different classes and for the training and test sets.
2) In L189, a more detailed description of what has been considered in the multiple tests, i.e., whether and what hyperparameters have been varied in case.
3) In L226, a small description of the two experiments before going into detail would be advised.
4) A summary table for experiment n.2 similar to what presented in Table 1 would be recommended.
Author Response
Dear Editor and reviewer #1:
Please find attached, the revised manuscript (Manuscript ID applsci-1308441):
Deep-Learning Applications in Geosciences: Insights into Ichnological Analysis
By: Korhan Ayranci*, Isa E. Yildirim, Umair bin Waheed and James A. MacEachern
We thank the reviewers and editor for their thoughtful comments aimed to improve the quality of the manuscript. We have carefully considered their second round of comments and suggestions, and revised our manuscript accordingly. The following summarizes our responses to the reviewers and editor comments (in red).
Response to comments from reviewer #1
1) In Sec.2.1 a good qualitative description of the dataset from ichnological perspective is provided; however, the reivewer would suggest to illustrate also the number of images considered for the different categories, whether grayscale or colour, and in which proportions for the different classes and for the training and test sets.
Thank you for pointing this out. We have now added the requested information: color images and percentage values of test and training images. Other information on number of test and training data have been given in lines 153-155.
2)In L189, a more detailed description of what has been considered in the multiple tests, i.e., whether and what hyperparameters have been varied in case.
We have added additional text to address this point.
3) In L226, a small description of the two experiments before going into detail would be advised.
Done. (Line 211-213)
4) A summary table for experiment n.2 similar to what presented in Table 1 would be recommended.
Although we previously included experiment #2 results in the same table 1, we found this comment very useful: we have now divided the table 1 into two and provided experiment results separately. Thank you for this comment.
Sincerely,
Korhan Ayranci

Reviewer 2 Report
As far as I am concerned, the paper must not be further modified
Author Response
Dear Editor and reviewer #2:
Please find attached, the revised manuscript (Manuscript ID applsci-1308441):
Deep-Learning Applications in Geosciences: Insights into Ichnological Analysis
By: Korhan Ayranci*, Isa E. Yildirim, Umair bin Waheed and James A. MacEachern
We thank the reviewer #2 for his/her thoughtful comments aimed to improve the quality of the manuscript.
Sincerely,
Korhan Ayranci
